# Reconfigurable Terahertz Metamaterials Based on the Refractive Index Change of Epitaxial Vanadium Dioxide Films Across the Metal–Insulator Transition

**DOI:** 10.3390/nano15060439

**Published:** 2025-03-13

**Authors:** Chang Lu, Weizheng Liang

**Affiliations:** 1Department of Electronic Communication and Technology, Shenzhen Institute of Information Technology, Shenzhen 518029, China; luchang2010@foxmail.com; 2State Key Laboratory of Featured Metal Materials and Life-Cycle Safety for Composite Structures, School of Resources, Environment and Materials, Guangxi University, Nanning 530004, China

**Keywords:** metamaterial, terahertz, phase-change material, vanadium dioxide, dielectric change, carrier confinement effect, percolation process

## Abstract

The intrinsic metal–insulator transition (MIT) of VO_2_ films near room temperature presents significant potential for reconfigurable metamaterials in the terahertz (THz) frequency range. While previous designs primarily focused on changes in electrical conductivity across the MIT, the accompanying dielectric changes due to the mesoscopic carrier confinement effect have been largely unexplored. In this study, we integrate asymmetric split-ring resonators on 35 nm epitaxial VO_2_ film and identify a “dielectric window” at the early stages of the MIT. This is characterized by a redshift in the resonant frequency without a significant degradation in the resonant quality. This phenomenon is attributed to an inhomogeneous phase transition in the VO_2_ film, which induces a purely dielectric change at the onset of the MIT, while the electrical conductivity transition occurs later, slightly above the percolation threshold. Our findings provide deeper insights into the THz properties of VO_2_ films and pave the way for dielectric-based, VO_2_ hybrid reconfigurable metamaterials.

## 1. Introduction

Engineering the properties of propagating electromagnetic waves is critical for applications in fields such as wireless communication, security imaging, and material spectroscopy [1,2,3,4,5]. Metamaterials composed of metal resonators and phase-change materials (PCMs) have emerged as an effective strategy, combining the benefits of subwavelength metallic structures with the active material properties of PCMs [6,7]. Various resonances, including dipole [8,9,10], LC (inductive-capacitive)-type [11,12], Fano-type [13], and EIT (electromagnetically induced transparency) [14] have been coupled with PCMs. In the THz regime, four types of PCMs are used: those exhibiting significant electrical conductivity changes, such as VO_2_, which enable the connection and disconnection between capacitive arms; dielectric-type PCMs, like STO [15], that tune the resonant frequency by changing the refractive index (RI); liquid crystals-hybrid metamaterial allows for tunable absorption coefficient and rewritable data storage by switching the liquid crystal alignment [16,17,18]; and those behaving superconducting transition, such as YBCO [12] and Nb [19], are less frequently studied due to the ultra-low phase-transition temperatures.

Among these PCMs, VO_2_ has garnered the most attention due to its near-room-temperature metal–insulator transition (MIT), large variant in electrical conductivity by several orders of magnitude, and simple fabrication process [7,20,21,22,23,24,25,26]. Most VO_2_-based reconfigurable metamaterials focus on their electrical conductivity change. Various THz applications, such as all-optical memories [13,27], imaging [10], tunable phase shifters [28], frequency selectors [29], and polarization converters [30], have been developed leveraging this effect. However, it was also observed that at the early stage of MIT, a hybrid structure of metal resonators and VO_2_ film enables frequency shifts of resonant dips, indicating a dielectric-type modulation effect. This dielectric change in VO_2_ film, as explained by the Drude–Smith model [31,32], is attributed to carrier confinement in nanostructures, where conduction electrons are confined and behave as diffused currents. A deeper understanding of this mechanism can help optimize the effect and achieve dielectric-type modulation in VO_2_-based THz metamaterials.

In this study, we demonstrate a dielectric-type modulation effect in a metamaterial consisting of asymmetric split-ring resonators (ASRRs) deposited on a 35 nm thick epitaxial VO_2_ film. Under thermal stimuli, a dielectric window is observed at temperatures below the electrical conductivity threshold, where the transmission spectra of ASRRs/VO_2_ show a clear redshift in resonant frequency without obvious degeneration in resonant intensity and quality factor (Q-factor). Finite element analysis of the results suggests an RI increase in VO_2_ from 3 to 27.5 at this temperature window. To clarify, we provide a microscopic explanation of the dielectric window based on a modified Drude–Smith model, which assigns the dielectric window to the carrier confinement effect at early MIT stages. We hope our findings offer a better understanding of the THz properties of epitaxial VO_2_ films and inspire new ideas for reconfigurable THz metamaterials hybridized with PCMs.

## 2. Fabrication and Characterization

Figure 1a schematically shows the design of the reconfigurable metamaterial, which consists of a metal resonator (Au/Ni, 200/10 nm), a VO_2_ film (35 nm), and a sapphire substrate, with an inset displaying a microscopic photograph of the metamaterial. The metal resonator is made up of a pair of asymmetric split-ring resonators (ASRRs), and the design parameters are listed in Figure 1b. The structural parameters are as follows: p = 80 μm, s = 45 μm, lw = 6 μm, h = 70 μm, and g = 6 μm. The asymmetric geometry of metal resonators gives rise to high-Q resonances that is sensitive to the dielectric change in the VO_2_ film. The epitaxial VO_2_ films were deposited onto the m-cut sapphire substrate via polymer deposition method. The epitaxial relationship between VO_2_ film and substrate is (-402) VO_2_ || AL_2_O_3_ (10-10), as confirmed by X-ray diffraction (XRD) patterns, which are provided in the Appendix A.

The temperature-dependent DC conductivity, presented in Figure 1c and measured using a four-point probe method, shows a sharp change of over four orders of magnitude, demonstrating the excellent quality of the MIT in the epitaxial VO_2_ film. High-resolution DC conductivity ($\sigma_{DC}$) measurements allow for precise identification of the MIT process. The derivative function of $\log\left(\sigma_{DC}\right)$ is plotted in Figure 1d and fitted to a Gaussian distribution [33], $N\left(T_{c},\;{\Delta T}^{2}\right)$, where $T_{c}$ is the critical temperature, and 2ΔT reflects the temperature window of the MIT. The Gaussian fitting to the DC experimental results yields $T_{c}=60.4\;{}^{\circ}C$ and ΔT=3.8 °C.

## 3. Results and Discussion

### 3.1. Simulations for Transmission Spectra

In this study, the transmission spectra of the designed ASRR/VO_2_ metamaterial were numerically investigated using the finite element analysis (FEA) method. The RI of the sapphire substrate, treated as a lossless dielectric material, was set to 3.45, and the ASRRs were modeled as perfect electric conductors. Importantly, the electrical conductivity ($\sigma_{THz}$) and refractive index ($n_{THz}$) of VO_2_ film at THz frequencies were treated as independent parameters, unlike in some works where an equation linking them is constructed within the Drude model framework. As demonstrated in our previous research, the relationship between the electrical and dielectric properties of VO_2_ across the MIT is complex and extends beyond the standard Drude model [34]. Specifically, setting $\sigma_{THz}$ to range from 0.1 to 200 S/cm, and $n_{THz}$ from 3 to 35 is sufficient to cover the actual $n_{THz}$ and $\sigma_{THz}$ changes in VO_2_ film below and near *T*_c_ (revealed by the experiment results in Appendix A).

The transmission spectra of ASRR/VO_2_ under a plane wave (TE mode, E||x-axis) are shown in Figure 2a. With $\sigma_{THz}=0.1$ S/cm and $n_{THz}=3$ (black line in Figure 2a), two narrow resonant dips appear at 0.47 THz (R1) and 0.71 THz (R2), accompanied by an EIT peak at 0.63 THz. To further explore the resonant mechanism, the electric field (Figure 2b), current direction (white arrow in Figure 2c), and magnetic field distributions (Figure 2c) at 0.47, 0.63, and 0.71 THz are examined. Initially, the electric field along the inductive metal arms generates LC-type electric dipole (ED) resonance, with the electric field at 0.47 THz confined near the capacitive arms (Figure 2b(i)). However, as displacement of two capacitive split gaps away from the central vertical axis, the right–left mirror symmetry of the ASRR pair is broken. This results in a pronounced head-to-tail closed current distribution on the upper and lower semi-circles of the ASRR pair (Figure 2c(ii)), creating out-of-plane magnetic dipoles (MDs). The MDs trap energy in the local field and induce destructive interference in the far field, producing the EIT peak at 0.63 THz and R2 dip at 0.71 THz. The R2 resonance is the so-called dark mode resonance, as it originates from the destructive interference between the in-plane ED and out-of-plane MDs. It should be noticed that field enhancement at EIT and R2 is more intense than that at R1, as shown in Figure 2b. This gives rise to the better dielectric sensitivity of R2 compared with R1 due to enhanced light-matter interactions. In conclusion, the asymmetric geometry of the ASRR pair leads to the dark mode and EIT peak, splitting the broad bright mode into two narrower resonant dips, providing a proper platform for the modulation effect induced by MIT.

Figure 2a shows the evolution of the transmission spectra as $n_{THz}$ increases from 3 to 35 ($\sigma_{THz}=0.1$ S/cm). It is evident that changes in $n_{THz}$ can induce a significant frequency shift of the resonances in the transmission spectra (Figure 2a). To quantify and compare the resonant properties, we fitted the transmittance spectra using Lorentzian resonance functions (R1 and R2) on a straight-line background. The resonant frequency, intensity (normalized to the intensity at $n_{THz}$ = 3), and the Q-factor were extracted from the fitting results. Representative fitting processes both for numerical and experiment results are provided in Appendix A. Analysis of the resonant frequency reveals a linear relationship with ${n_{THz}}^{2}$ (Figure 2d(i)). Thus, we defined the change rate of resonant frequency with respect to ${n_{THz}}^{2}$ as: sensitivity= $\Delta f/\Delta \left({n_{THz}}^{2}\right)$, where Δf is the frequency shift, and $\Delta {n_{THz}}^{2}={n_{THz}}^{2}-{n_{0}}^{2}$ represents the dielectric constant change of VO_2_ film. As shown in Figure 2d(i), the sensitivity for R1 is −52 MHz/RIU^2^, and for R2 is −78 MHz/RIU^2^ (where RIU stands for refractive index unit). The resonant intensity and Q-factor (Figure 2d(ii,iii)) show a negligible change for R1 as $n_{THz}$ increases from 3 to 35, while for R2, the Q-factor decreases from 17.4 to 9.1, and normalized intensity increases from 1.0 to 1.6.

### 3.2. ASSR/VO_2_ Spectra Influenced by Conductivity Change

In addition to the dielectric modulation effect, the impact of electrical conductivity changes on the ASRRs/VO_2_ spectra is also non-negligible. A representative case is shown in Figure 3a with $\sigma_{THz}$ ranging from 0.1 to 1000 S/cm at $n_{THz}=20$. As $\sigma_{THz}$ increases from 0.1 to 500 S/cm, R1 and R2 progressively weaken, along with the EIT peak at 0.61 THz. When $\sigma_{THz}$ exceeds 500 S/cm, R2 is no longer observed, and R1 becomes significantly broadened. To investigate the underlying mechanism, the electric field distributions at 0.68 THz for $\sigma_{THz}$ values of 0.1, 100, and 500 S/cm are shown in Figure 3b. It is evident that field enhancement at these frequencies is significantly influenced by $\sigma_{THz}$, as increased electrical conductivity in the VO_2_ film facilitates charge transfer between the capacitive gaps.

To illustrate the impact of conductivity change, the resonant frequency shift, intensity (normalized), and Q-factor of R1 and R2 were extracted from Figure 3a based on Lorentz fitting, and plotted as functions of $\sigma_{THz}$ in Figure 3c(i–iii). The resonant intensity (Figure 3c(ii)) and the Q-factor (Figure 3c(iii)) exhibit a negative relationship with increasing $\sigma_{THz}$, and the effect of $\sigma_{THz}$ is more pronounced on R2 than on R1. The frequency shifts (Figure 3c(i)) of R1 and R2 approaches zero in the range 0.1 < $\sigma_{THz}$ < 200 S/cm, and becomes obvious after $\sigma_{THz}$ exceeds 400 S/cm. Precisely, the maximum frequency shift (Δf) in the range 0.1 < $\sigma_{THz}$ < 200 S/cm is ~1 GHz for R1, and ~2.5 GHz for R2, as shown in the inset of Figure 3c(i).

To further evaluate the influence of $\sigma_{THz}$ on resonant properties, transmission spectra when $\sigma_{THz}$ changes from 0.1 to 200 S/cm at $n_{THz}=3,\;10,\;20,\;25,\;30,and\;35$ are calculated (these spectra are provided in Appendix A). The resonant frequencies of R1 and R2 from these simulations are summarized in Figure 3d as functions of $\sigma_{THz}$, with $f_{0}$ (at $\sigma_{THz}$ = 0.1 S/cm) marked by dotted lines. It could be observed that, for a fixed $n_{THz}$, the resonant frequency exhibits only minor shifts around $f_{0}$. The maximum frequency deviation observed in the range 0 < $\sigma_{THz}$ < 200 S/cm is 1.7 GHz for R1 (at $n_{THz}$ = 10), and 3.3 GHz for R2 (at $n_{THz}$ = 30). These shifts correspond to approximately 4.8% and 5.9% of the experimental frequency shifts (35 GHz for R1 and 54 GHz for R2, discussed in Section 3.3), respectively. It suggests that the conductivity change at temperatures below *T*_c_ is not the dominant reason for the observed resonant frequency shift in this work.

In Figure 3e, resonant frequencies are plotted as functions of ${n_{THz}}^{2}$ under different electrical conductivity. It is shown that the linear relationship between ${n_{THz}}^{2}$ and the resonant frequency is unaffected by the value of $\sigma_{THz}$. The dielectric sensitivity extracted from the slope of the linear fitting is plotted in Figure 3f as a function of $\sigma_{THz}$. The results show that the sensitivity of R1 and R2 is nearly constant over 0~200 S/cm, showing a mean value at 51 MHz/RIU^2^ for R1 (with a standard error of 0.24 MHz/RIU^2^), and the mean value for R2 is 78 MHz/RIU^2^ (with stand error 0.33 MHz/RIU^2^).

### 3.3. Experimental Results

From the numerical results, we demonstrated that at temperatures below *T*_c_, the altered dielectric constant of VO_2_ film can modulate the resonant frequency of ASRRs, while changes in electrical conductivity have negligible effect. This is further supported by the experimental results shown in Figure 4a, where the transmission spectra of ASRRs/VO_2_ are presented as a function of temperature in the heating process. The spectra were measured using fiber-coupled THz time-domain spectroscopy (representative time-domain signal provided in Appendix A), with the sample placed on a high-resolution temperature controller. As depicted in Figure 4a, at temperatures ranging from 54 °C to 60 °C (below $T_{c}$), R1 and R2 maintain narrow linewidths, and their center frequencies shift redward as the temperature increases. Upon reaching $T_{c}$ (60–61 °C), the linewidths of R1 and R2 broaden significantly, and their resonant intensities decrease rapidly. Above $T_{c}$ (>61 °C), the transmission spectra become featureless, indicating that resonance behavior has completely vanished. The transmission spectra of ASRR/VO_2_ during the cooling process are provided in Appendix A. Specifically, as the temperature decreases and VO_2_ film transitions from a metallic state back to an insulating state, the resonances R1 and R2 re-emerge (from 55 °C to 53 °C) and then blue-shift (from 53 °C to 50 °C). This behavior is consistent with the expected hysteresis effect of VO_2_ during the MIT.

The resonant frequency, intensity, and Q-factor of R1 and R2 as a function of temperature are shown in Figure 4b,c both in the heating and cooling processes. The resonant properties in Figure 4b,c further illustrate that the hysteresis loop width of the modulation effect is approximately 7 °C, similar to the loop width (~7.6 °C) measured in DC conductivity. In the heating process, the frequency shift of the resonances increases with temperature, reaching a maximum just above $T_{c}$ (Figure 4b(i),c(i)). The maximum frequency shift (Δf) at 61 °C for R1 is 70 GHz, while for R2, it is 105 GHz. Additionally, the intensities of R1 and R2 at 61 °C are reduced to 51% and 59%, respectively, compared to their values at 50 °C. The Q-factor of R1 remains constant at approximately 2.1, while the Q-factor of R2 decreases from 7.0 to 4.0 as the frequency shift reaches its maximum.

However, for resonator applications, significant reductions in resonant properties should be avoided. In this case, it is interesting to note that R1 and R2 maintain good resonant properties at temperatures below $T_{c}$. Specifically, when comparing the resonant properties at 60 °C to those at 54 °C, the intensity of R1 decreases by 33%, the intensity of R2 decreases only by 8%; the Q-factor of R1 remains nearly constant at around 2.0, and the Q-factor of R2 decreases slightly from 7.0 to 5.0. Therefore, the maximum frequency shifts for ASRR/VO_2_, in cases of high resonance quality, are 35 GHz for R1 and 56 GHz for R2.

In conclusion, we experimentally demonstrate that ASRR/VO_2_ structure enables two kinds of modulation effects based on temperature control—the dielectric-induced resonant frequency shift at temperatures below the *T*_c_ and conductivity transition-induced transmission decrease above the *T*_c_. This opens new possibilities for THz applications in need of multistate field control, such as optical memories, active modulators, and THz imaging.

### 3.4. THz Properties of VO_2_ Film

As discussed in Section 3.2, a conductivity change from 0.1 to 200 S/cm should not be the dominant reason for the measured frequency shifts of R1 and R2. This suggests that the measured frequency shift in Figure 4 can be assigned to a dielectric change of VO_2_ film. The corresponding $n_{THz}$ change can be extracted based on the dielectric sensitivity of R1 and R2:(1)$$n_{THz}={\left[sensitivity*\Delta f+{n_{THz}}^{2}\left(RT\right)\right]}^{0.5}$$
where $n_{THz}\left(RT\right)$ are the refractive index of the VO_2_ film at room temperature. The $n_{THz}$ (Figure 5a) acquired from ASRRs/VO_2_ spectra are limited to 61 °C; beyond this temperature, the resonant properties become difficult to discern. Two groups of $n_{THz}$ are presented in Figure 5a, acquired from the frequency shift of R1 (original at 0.47 THz) and R2 (original at 0.71 THz), respectively. To better understand the THz properties across the entire MIT, we present the $n_{THz}$ (Figure 5b) and $\sigma_{THz}$ (Figure 5c) derived from bare 35 nm VO_2_ film (detailed experimental methods and calculations are provided in Appendix A). Despite the limited detection range, the $n_{THz}$ acquired from the ASRRs/VO_2_ spectra show better resolution and coherence compared to those directly obtained from the bare VO_2_ film. In detail, when the THz pulse transmits through a bare VO_2_ film, conductivity changes directly affect the amplitude of the transmitted signal, while variations in the refractive index give rise to time delays. However, due to the small thickness of VO_2_ film (35 nm), dielectric change in VO_2_ causes only a tiny time delay. In this scenario, the ASRR/VO_2_ structure with strong electric field enhancement at R1 and R2, enhances the light-matter interaction between THz pulse and VO_2_ film, providing higher sensitivity to dielectric change compared with bare VO_2_ film.

Compared the temperature dependence of $n_{THz}$ and $\sigma_{THz}$, a dielectric window (55–60 °C) is formed (green area in Figure 5a), where the VO_2_ film undergoes a dielectric change while suppressing the electrical conductivity change. As shown in Figure 5a, $n_{THz}$ at around 0.47 THz (acquired from R1) and 0.71 THz (from R2) begins to increase at 55 °C, whereas $\sigma_{THz}$ exhibits characteristic percolation behavior, remaining low (<100 S/cm) at 55–60 °C, and showing a noticeable increase only after the critical threshold of 61 °C has been reached.

### 3.5. THz Properties Influenced by Percolation Effect

Primarily, the dielectric window at the early stage of the MIT is attributed to the percolation effect in electrical conductivity. To clarify this, the Bruggeman Effective Medium Approximation (EMA), which models a randomly percolating system consisting of polarized powders and a homogeneous matrix, is used to describe the THz properties of the VO_2_ film. The most commonly used formula is given by [31]:(2)$$p_{m}\frac{n_{m}-n_{eff}}{n_{m}+\left(g*-1\right)n_{eff}}+\left(1-p_{m}\right)\frac{n_{i}-n_{eff}}{n_{i\;}+\left(g*-1\right)n_{eff}}=0$$
where $p_{m}$ is the volume fraction of metallic domains, g* is a parameter that reflects the percolation threshold, and $n_{i}$, $n_{m}$, and $n_{eff}$ are the corresponding refractive index at insulating, metallic, and inhomogenous states, respectively. The metallic phase volume fraction $p_{m}$ during a quasi-static heating process can be modeled by integrating the Gaussian distribution function [33]:(3)$$p_{m}\left(T\right)=\int_{0}^{T}\frac{1}{\sqrt{2\pi }\Delta T}exp\frac{-{\left(T-T_{c}\right)}^{2}}{2\Delta T^{2}}dT$$
where $T_{c}=60.4\;{}^{\circ}C$ and ΔT=3.8 °C are derived from DC conductivity (Section 2).

To investigate the percolation effect in VO_2_ film, $n_{THz}$, $\sigma_{THz}$, and $\sigma_{DC}$ are plotted (Figure 6a) as a function of $p_{m}$ and fitted to Equation (2). For the RI change, we set $n_{i}=3,\;n_{m}=45$, and $n_{eff}$ to the RI derived from the ASRR/VO_2_ spectra, and find that the percolation threshold g*=0. A noticeable non-zero percolation threshold is observed for the electrical conductivities at THz (Figure 6a(ii)) and DC (Figure 6a(iii)). As a result, the EMA with $\sigma_{i}=4\;S/cm,\;\sigma_{m}=3600\;S/cm$, and g*=0.70 fits the THz conductivity, while $\sigma_{i}=0.2\;S/cm,\;{\;\sigma }_{m}=3600\;S/cm,$ and g*=0.65 fit the DC conductivity. The critical percolation threshold g* for both DC and THz conductivity, around 0.65–0.70, corresponds to the volume fraction of a 2D percolation system and is consistent with the thin film geometry of VO_2_. Therefore, by modeling the experimental results with the EMA, it is found that the refractive index change in VO_2_ film shows a non-threshold behavior, while both DC and THz conductivity undergo percolation transitions at $p_{m}\approx 0.65$ to 0.70. Thus, the dielectric window formed at the early stage of the MIT is attributed to the differing percolation effects in the physical properties.

### 3.6. Microscopic Origin of Dielectric Window

While the EMA is a phenomenological model rather than a physical mechanism, the Drude–Smith model and its derivatives offer a more microscopic explanation of electron dynamics in nanostructured thin films [35]. Here, we provide an explanation using a modified Drude–Smith model developed by Cocker et al., based on Monte Carlo simulations, supporting the existence of a dielectric window at the early stage of the MIT from a microscopic perspective [36]. In this model, electron dynamics in the VO_2_ film are primarily influenced by the ratio of the metallic domain linear dimension ($L_{d}$) to the electron diffusion distance ($v_{th}\tau$), where $v_{th}$ is the thermal velocity and τ is the intrinsic scattering time. Assuming a 100% back-scattering probability at domain boundaries, the modified Drude–Smith model gives the following expression for conductivity [36]:(4)$$\tilde{\sigma }(\omega )=\frac{Ne^{2}\tau^{\prime }/m*}{1-i\omega \tau^{\prime }}(1-\frac{1}{1-i\omega t_{diff}})$$
where $\tilde{\sigma }=\sigma_{1}+{i\sigma }_{2}$ is the complex electrical conductivity, N is the carrier density, e is the elementary charge, m* is the effective mass of charge carriers, where τ′ is an effective scattering time combining the intrinsic scattering and scattering at domain boundaries (1/$\tau^{\prime }=1/\tau +1/\tau_{B}$, with $\tau_{B}=L_{d}/2v_{th}$). $t_{diff}={L_{d}}^{2}/(12D\prime )$ is a diffusion parameter following Einstein’s relation ($D^{\prime }=\tau^{\prime }k_{B}T/m$*). The THz electrical conductivity discussed in the previous section corresponds to the real part of $\sigma_{1}$, i.e., $\sigma_{THz}=\sigma_{1}$, while $\sigma_{2}$ is responsible for the RI change associated with the MIT, as described by: ${n_{THz}}^{2}=\varepsilon_{\infty }-\sigma_{2}/\omega \varepsilon_{0}$.

Figure 6b shows the evolution of $\sigma_{1}$ and $\sigma_{2}$ (at 0.5 and 0.7 THz) as a function of $L_{d}/v_{th}\tau$, modeled by Equation (4). The evolution is divided into three stages, (i) $L_{d}/v_{th}\tau <2$, corresponding to complete confinement, (ii) $2{<L}_{d}/v_{th}\tau <10$, for partial confinement, (iii) $L_{d}/v_{th}\tau >10$, for released confinement. Previous near-field microscopy studies have shown that the size of the nucleated metallic domains in epitaxial VO_2_ films is on the order of ~10 nm [37,38,39,40], which is comparable to the electron diffusion length [32,41] and corresponding to the complete confinement scenario depicted in Figure 6b(i). In detail, at the early stage of the MIT, the electrons are fully confined within the isolated metallic domains, resulting in the diffusive restoring current (originates from intrinsic scattering and domain boundary scattering). The resulting zero $\sigma_{1}$ and negative $\sigma_{2}$ is responsible for the dielectric window below the percolation threshold.

The model also predicts a significant increase in $\sigma_{1}$ as $L_{d}/v_{th}\tau$ increases from 2 to 10 (Figure 6b(ii)), with $\sigma_{1}$ reaching its maximum when $L_{d}/v_{th}\tau >30$ (Figure 6b(iii)). This aligns with the observed abrupt increase in electrical conductivity well above the percolation threshold. Thus, both the modified Drude–Smith model and the EMA consistently support the existence of a dielectric window in VO_2_ films at THz frequencies below the percolation threshold.

## 4. Conclusions

By measuring the transmission spectra of ASRRs/VO_2_, we demonstrated the existence of a “dielectric window” in an epitaxial VO_2_ film at the early stages of MIT, where the VO_2_ film undergoes dielectric change and suppresses the electrical conductivity change. This mechanism allows the resonant frequency of ASRRs/VO_2_ to redshift at 54~60 °C without significant degradation in the resonant intensity or the Q-factor. The maximum frequency shift, occurring well below the electrical conductivity threshold, is ~35 GHz for R1 ($f_{0}$ at 0.47 THz) and ~56 GHz for R2 ($f_{0}$ at 0.71 THz), corresponding to an increase in the refractive index $n_{\mathrm{THz}}$ of the VO_2_ film from 3 to 27. Our findings not only highlight the potential for dielectric-type modulation of resonant frequency enabled by VO_2_ films but also provide new insights into the MIT mechanisms in epitaxial VO_2_ films.

## Figures and Tables

**Figure 1 nanomaterials-15-00439-f001:**
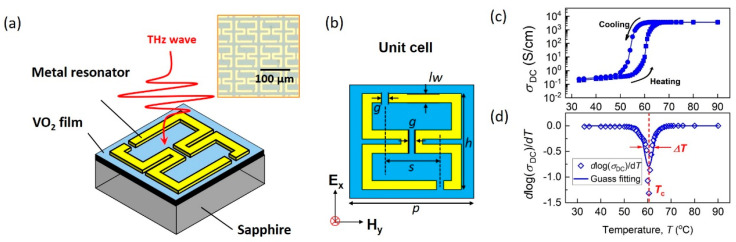
Schematic of the reconfigurable metamaterial and its characteristics: (**a**) the metamaterial structure consists of asymmetric split ring resonators (ASRRs) made from Au/Ni, a VO_2_ film, and a sapphire substrate, with an inset showing a microscopic image of the metamaterial; (**b**) structural parameters of the ASRRs, including p = 80 μm, s = 45 μm, lw = 6 μm, h = 70 μm, and g = 6 μm; (**c**) temperature-dependent DC conductivity of the VO_2_ film, showing a sharp transition; (**d**) derivative of the $\log\left(\sigma_{DC}\right),$ fitted to a Gaussian distribution $N\left(T_{c},{\Delta T}^{2}\right)$, with the critical temperature $T_{c}$ and the temperature window half-width ΔT.

**Figure 2 nanomaterials-15-00439-f002:**
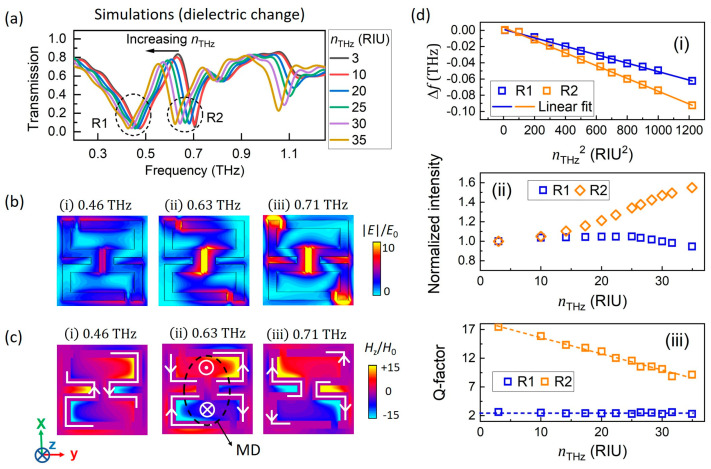
Transmission spectra over dielectric change and resonant mechanism of ASRR/VO_2_: (**a**) transmission spectra under TE plane wave (E||*x*-axis) as a function of variant $n_{THz}$; (**b**) electric field (magnitude) distribution at 0.47 (R1), 0.63, and 0.71 (R2) THz; (**c**) current direction (white arrow) and magnetic field distribution at 0.47, 0.63, and 0.71 THz. An out-of-plane magnetic dipole (MD) at 0.63 THz is circled in black dash line; (**d**) resonant properties extracted from Lorentz fitting are shown in (**i**) frequency, (**ii**) intensity (normalized to intensity at $n_{THz}$ = 3), and (**iii**) Q-factor as a function of $n_{THz}$.

**Figure 3 nanomaterials-15-00439-f003:**
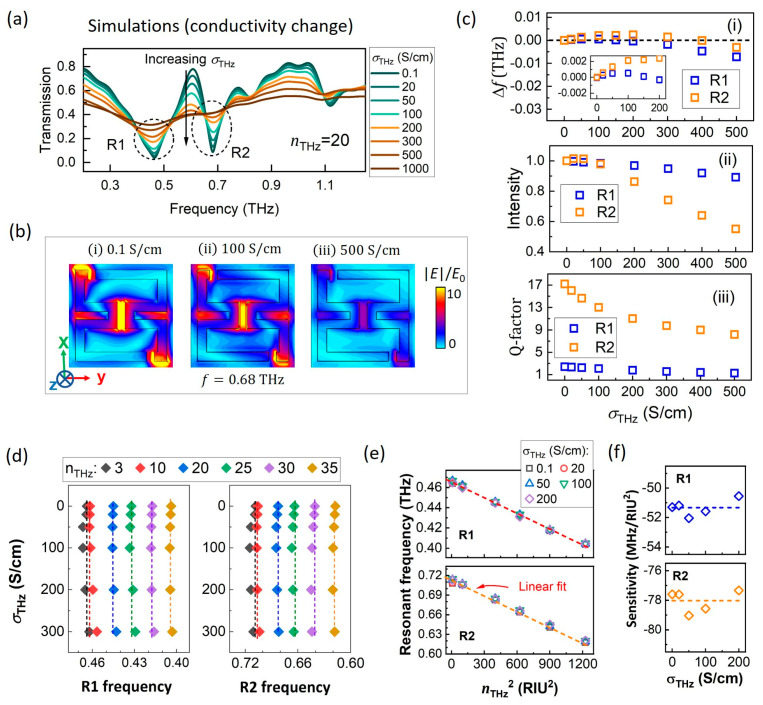
Spectral change induced by conductivity transition: (**a**) transmission spectra of ASRR/VO_2_ at $n_{THz}=20$, and $\sigma_{THz}$ = 0.1–1000 S/cm under TE plane wave (E||*x*-axis); (**b**) electric field (magnitude) distribution at 0.68 THz for $\sigma_{THz}$ = 0.1, 100, and 500 S/cm; (**c**) resonant properties extracted from Lorentz fitting are shown for (**i**) frequency, (**ii**) intensity (normalized to intensity at $n_{THz}$ = 3, $\sigma_{THz}$ =0.1 S/cm), and (**iii**) Q-factor; (**d**) resonant frequency of R1 and R2 as functions of $\sigma_{THz}$, corresponding to simulations in Appendix A; (**e**) resonant frequency plotted as function of ${n_{THz}}^{2}$. The dielectric sensitivity extracted from the slope of linear fitting is shown in (**f**).

**Figure 4 nanomaterials-15-00439-f004:**
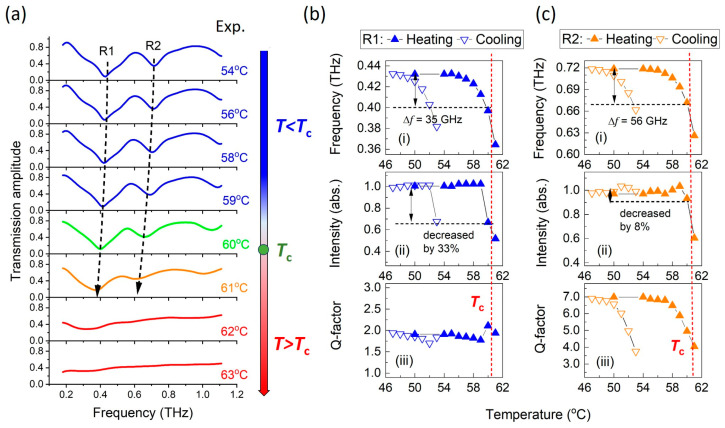
Measured temperature-dependent transmission spectra and resonant properties of ASRRs/VO_2_: (**a**) transmission spectra of ASRR/VO_2_ as a function of temperature in the heating process. The spectra in the cooling process are provided in Appendix A; (**b**) and (**c**) show the (**i**) resonant frequency shift, (**ii**) intensity (normalized to intensity at room temperature), and (**iii**) Q-factor of R1 and R2 as functions of temperature in the heating process (solid triangles) and cooling process (open triangles).

**Figure 5 nanomaterials-15-00439-f005:**
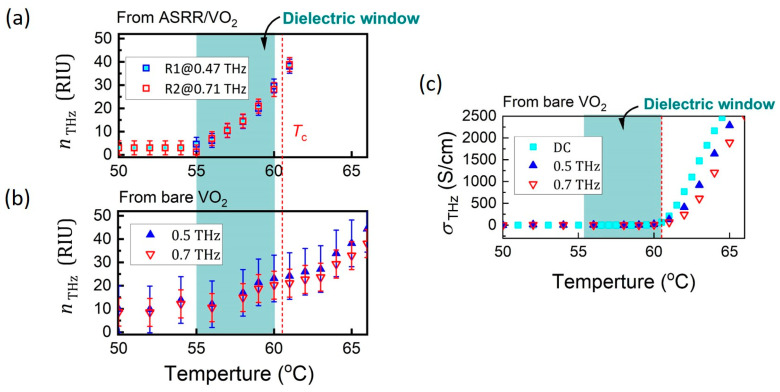
THz Properties of VO_2_ film in the heating process. Temperature-dependent refractive index $n_{THz}$ extracted from (**a**) the dielectric sensitivity of R1 and R2 in ASRRs/VO_2_ spectra and (**b**) bare VO_2_ film. (**c**) Electrical conductivity $\sigma_{THz}$ as a function of temperature extracted from bare VO_2_ film.

**Figure 6 nanomaterials-15-00439-f006:**
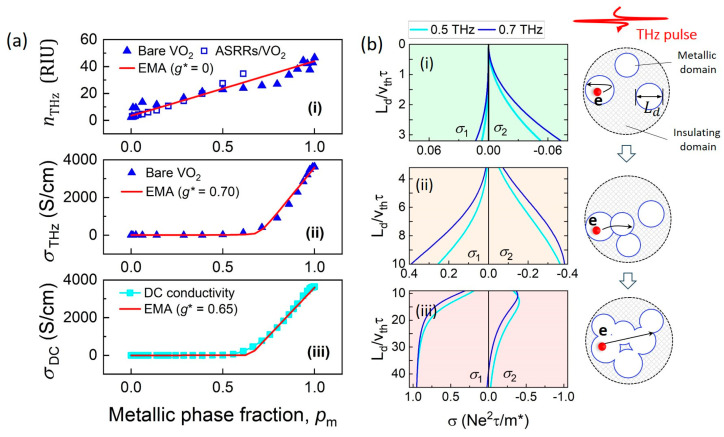
Mechanisms underlying the dielectric window: (**a**) percolation effects in VO_2_ film. Refractive index (**i**), THz conductivity (**ii**), and DC conductivity (**iii**) as a function of metallic domain volume fraction $p_{m}$; (**b**) evolution of $\sigma_{1}$ and $\sigma_{2}$(at 0.5 and 0.7 THz) as a function of $L_{d}$/$v_{th}\tau$ according to modified Drude–Smith model (Equation (4)) at three representative stages: (**i**) $L_{d}$/$v_{th}\tau <2$ corresponding to complete carrier confinement, (**ii**) $3{<L}_{d}/v_{th}\tau <10$, for partial confinement, (**iii**) $L_{d}/v_{th}\tau >10$ for released confinement. The schematics on the right illustrate electron dynamics at these three stages.

## Data Availability

Data is contained within the article.

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
