# Peer review of "Reconfigurable Terahertz Metamaterials Based on the Refractive Index Change of Epitaxial Vanadium Dioxide Films Across the Metal–Insulator Transition"

_nanomaterials, 2025, doi:10.3390/nano15060439_

Round 1
Reviewer 1 Report
Comments and Suggestions for Authors
The manuscript entitled “Reconfigurable Terahertz Metamaterials Based on the Refractive Index Change of Epitaxial Vanadium Dioxide Films Across the Metal-Insulator Transition” by Chang L. and Weizheng L. studies the dielectric-type change of ASRR/VOâ‚‚ metamaterials. The authors performed both theoretical simulations and experimental measurements of the transmission spectra for this structure and observed the dielectric window during the VOâ‚‚ MIT, which could potentially shift the resonator frequency without significantly affecting the resonant quality. Overall, this phase transition feature is intriguing and could be beneficial for future THz metamaterials using VOâ‚‚ films. However, I have several concerns and comments regarding the simulation results and analysis presented in this manuscript. These issues should be addressed before I can further support the acceptance of this manuscript.
(1) In Sec. 3.1, the authors treat conductivity σTHz and the refractive index nTHz as independent parameters. Although it is reasonable to consider them as separate variables due to the complexity of the metal-insulator transition (MIT) in VOâ‚‚, they become less independent when VOâ‚‚ is near the metallic phase, where the Drude model is more applicable.
For the simulations in Fig. 2, the assumption of negligible THz conductivity with increasing is reasonable, given the dramatic conductivity change around the transition temperature, as shown in Fig. 1c. However, in Fig. 3, the simulated σTHz varies from 0.1 to 1000 S/cm. This range mostly corresponds to temperatures region close to Tc, where VOâ‚‚ is nearly metallic. In this case, assuming a fixed value nTHz = 3 may not be appropriate, as σTHz and nTHz are no longer independent. Consequently, the conclusion that “changes in σTHz have little effect on the resonant frequency” (Line 142) could be misleading.
(2) In Sec. 3.3, the authors use Eq. (1) to extract σTHz and nTHz. Did the authors obtain these values using sensitivity and decay rate from simulations? If so, the simulated results might not accurately represent the actual scenario, as discussed in the first comment. Additionally, the constant parameters could be problematic since the sensitivity and decay rate likely vary with temperature in the real case. Furthermore, why did the authors not use THz transmission spectra to extract the refractive index and conductivity from the ASRR/VOâ‚‚ structure, as they did for the bare VOâ‚‚ film? This would provide more direct evidence supporting the existence of the dielectric window.
(3) The modified Drude-Smith model in Sec. 3.5 seems to effectively describe the microscopic properties of VOâ‚‚ film during the MIT. Would it be possible to use this model to generate more realistic parameters σTHz and nTHz for the transmission spectra simulations in Sec. 3.1? If so, the confinement could be linked to the temperature and potentially fitted using experimental data. It can probably solve the issue in the first comment.
(4) The experiment focuses on transmission spectra changes during the heating process. However, how would the spectra behave during the cooling process? Given the hysteresis in the MIT of VOâ‚‚, as shown by DC measurements, this phenomenon might also influence the THz properties of the ASRR/VOâ‚‚ metamaterial. It would be beneficial to include data from the cooling process to provide a more comprehensive study.
(5) The resonant intensity figures in the manuscript are somewhat unclear. Were these figures derived from the raw THz spectra of ASRR/VOâ‚‚? If so, the authors should present the corresponding THz spectra, either in the main text or supplementary. Additionally, some representative time-domain traces are also needed.
Reviewer 2 Report
Comments and Suggestions for Authors
The authors proposed a dielectric-type modulation effect in a metasurface consisting of asymmetric split-ring resonators (ASRRs) deposited on a 35-nm-thick epitaxial VOâ‚‚ film. Importantly, under thermal stimuli, a dielectric window is observed at temperatures below the electrical conductivity threshold, where the transmission spectra of ASRRs/VOâ‚‚ show a clear redshift in resonant frequency without obvious degeneration in resonant intensity and quality factor (Q-factor). After thorough evaluation, I conclude that the manuscript is well thought out, presents an original concept, and has been carefully prepared. Nevertheless, I believe that a few minor corrections could enrich it somewhat. Therefore, I ask the authors to address the following issues:
- In the introduction, the authors mention that “in the THz regime, three types of PCMs are used”. However, they completely ignored liquid crystals (LCs) in this context, which have probably found the widest application in tuning terahertz metamaterials. It is worth mentioning the concepts of such active and hybrid THz metamaterials, e.g.: Appl. Phys. Lett. 106 (9): 092905 (2015); Symmetry 15.1 (2022): 103.
- What - from a physical point of view - was behind the design of this particular model of the metamaterial unit cell? Please justify it.
- In conclusion, it is worth devoting a separate paragraph to discuss possible applications of this type of active metamaterials in the THz range. What is their unique feature in terms of applications, what is their advantage over other materials?
Author Response
The authors proposed a dielectric-type modulation effect in a metasurface consisting of asymmetric split-ring resonators (ASRRs) deposited on a 35-nm-thick epitaxial VOâ‚‚ film. Importantly, under thermal stimuli, a dielectric window is observed at temperatures below the electrical conductivity threshold, where the transmission spectra of ASRRs/VOâ‚‚ show a clear redshift in resonant frequency without obvious degeneration in resonant intensity and quality factor (Q-factor). After thorough evaluation, I conclude that the manuscript is well thought out, presents an original concept, and has been carefully prepared. Nevertheless, I believe that a few minor corrections could enrich it somewhat. Therefore, I ask the authors to address the following issues:
Reply: Thanks for your valuable time and feedback. Below is a point-to-point response, with the corresponding revisions highlighted in the revised manuscript.
Comment 1: In the introduction, the authors mention that “in the THz regime, three types of PCMs are used”. However, they completely ignored liquid crystals (LCs) in this context, which have probably found the widest application in tuning terahertz metamaterials. It is worth mentioning the concepts of such active and hybrid THz metamaterials, e.g.: Appl. Phys. Lett. 106 (9): 092905 (2015); Symmetry 15.1 (2022): 103.
Response 1: We appreciate the reviewer's suggestion regarding providing a more comprehensive discussion on PCMs used in THz metamaterials, especially including those hybrids with LCs. Please find it in the introduction section:
“Liquid crystals-hybrid metamaterial allows for tunable absorption coefficient and rewritable data storage by switching the liquid crystal alignment16-18” (Page 1, lines 35-36)
This description corresponding references:
[16] Wuttig, M.; Yamada, N., Phase-change materials for rewriteable data storage. Nature Materials 2007, 6 (11), 824-832.
[17] Kowerdziej, R.; Jaroszewicz, L.; Olifierczuk, M.; Parka, J., Experimental study on terahertz metamaterial embedded in nematic liquid crystal. Applied Physics Letters 2015, 106 (9).
[18] Piccirillo, B.; Paparo, D.; Rubano, A.; Andreone, A.; Rossetti Conti, M.; Giove, D.; Vicuña-Hernández, V.; Koral, C.; Masullo, M. R.; Mettivier, G.; Opromolla, M.; Papari, G.; Passarelli, A.; Pesce, G.; Petrillo, V.; Piedipalumbo, E.; Ruijter, M.; Russo, P.; Serafini, L., Liquid Crystal-Based Geometric Phase-Enhanced Platform for Polarization and Wavefront Analysis Techniques with the Short-TeraHertz FEL Oscillator TerRa@BriXSinO. Symmetry 2023, 15 (1), 103.
Comment 2: What - from a physical point of view - was behind the design of this particular model of the metamaterial unit cell? Please justify it.
Response 2: We appreciate the reviewer's suggestion to provide a more precise discussion on the physical mechanism behind the optical unit cell. Please find it in the revised manuscript:
“The R2 resonance is the so-called dark mode resonance, as it originates from the destructive interference between the in-plane ED and out-of-plane MDs. It should be noticed that field enhancement at EIT and R2 is more intense than that at R1, as shown in Figure 2b. This gives rise to the better dielectric sensitivity of R2 compared with R1 due to enhanced light-matter interactions. In conclusion, the asymmetric geometry of the ASRR pair leads to the dark mode and EIT peak, splitting the broad bright mode to two narrower resonant dips, providing a proper platform for the modulation effect induced by MIT.” (Page 3, lines 117-123)
Comment 3: In conclusion, it is worth devoting a separate paragraph to discuss possible applications of this type of active metamaterials in the THz range. What is their unique feature in terms of applications, what is their advantage over other materials?
Response 3: We appreciate the reviewer's suggestion to provide a discussion on the feature and applications of this metamaterial. Please find it in the revised manuscript:
“In conclusion, we experimentally demonstrate that ASRR/VO2 structure allows for two kinds of modulation effect based on temperature control—the dielectric-induced resonant frequency shift at temperatures below the Tc and conductivity transition-induced transmission decrease above the Tc. This paves the way for THz applications in need of multistate field control, such as optical memories, active modulators, and THz imaging.” (Page 7, lines 238-242)

Round 2
Reviewer 1 Report
Comments and Suggestions for Authors
Thanks for the prompt revision by both authors. I have carefully reexamined the updated manuscript and noticed a significant improvement in the content. The revised figures and discussions have effectively addressed my concerns. I believe this version meets the required standards and should be accepted for publication in Nanomaterials.